# Sensor Selection in High-Dimensional Gaussian Trees with Nuisances

**Daniel Levine**
MIT LIDS
dlevine@mit.edu

**Jonathan P. How**
MIT LIDS
jhow@mit.edu

## Abstract

We consider the sensor selection problem on multivariate Gaussian distributions where only a *subset* of latent variables is of inferential interest. For pairs of vertices connected by a unique path in the graph, we show that there exist decompositions of nonlocal mutual information into local information measures that can be computed efficiently from the output of message passing algorithms. We integrate these decompositions into a computationally efficient greedy selector where the computational expense of quantification can be distributed across nodes in the network. Experimental results demonstrate the comparative efficiency of our algorithms for sensor selection in high-dimensional distributions. We additionally derive an online-computable performance bound based on augmentations of the relevant latent variable set that, when such a valid augmentation exists, is applicable for *any* distribution with nuisances.

## 1   Introduction

This paper addresses the problem of focused active inference: selecting a subset of observable random variables that is maximally informative with respect to a specified subset of latent random variables. The subset selection problem is motivated by the desire to reduce the overall cost of inference while providing greater inferential accuracy. For example, in the context of sensor networks, control of the data acquisition process can lead to lower energy expenses in terms of sensing, computation, and communication [1, 2].

In many inferential problems, the objective is to reduce uncertainty in only a *subset* of the unknown quantities, which are related to each other and to observations through a joint probability distribution that includes auxiliary variables called *nuisances*. On their own, nuisances are not of any extrinsic importance to the uncertainty reduction task and merely serve as intermediaries when describing statistical relationships, as encoded with the joint distribution, between variables. The structure in the joint can be represented parsimoniously with a probabilistic graphical model, often leading to efficient inference algorithms [3, 4, 5]. However, marginalization of nuisance variables is potentially expensive and can mar the very sparsity of the graphical model that permitted efficient inference. Therefore, we seek methods for selecting informative subsets of observations in graphical models that retain nuisance variables.

Two primary issues arise from the inclusion of nuisance variables in the problem. Observation random variables and relevant latent variables may be nonadjacent in the graphical model due to the interposition of nuisances between them, requiring the development of information measures that extend beyond adjacency (alternatively, *locality*) in the graph. More generally, the absence of certain conditional independencies, particularly between observations conditioned on the relevant latent variable set, means that one cannot directly apply the performance bounds associated with submodularity [6, 7, 8].

In an effort to pave the way for analyzing focused active inference on the class of general distributions, this paper specifically examines multivariate Gaussian distributions – which exhibit a number of properties amenable to analysis – and later specializes to Gaussian trees. This paper presents a decomposition of pairwise nonlocal mutual information (MI) measures on Gaussian graphs that permits efficient information valuation, e.g., to be used in a greedy selection. Both the valuation and subsequent selection may be distributed over nodes in the network, which can be of benefit for high-dimensional distributions and/or large-scale distributed sensor networks. It is also shown how an augmentation to the relevant set can lead to an online-computable performance bound for general distributions with nuisances.

The nonlocal MI decomposition extensively exploits properties of Gaussian distributions, Markov random fields, and Gaussian belief propagation (GaBP), which are reviewed in Section 2. The formal problem statement of focused active inference is stated in Section 3, along with an example that contrasts focused and unfocused selection. Section 4 presents pairwise nonlocal MI decompositions for scalar and vectoral Gaussian Markov random fields. Section 5 shows how to integrate pairwise nonlocal MI into a distributed greedy selection algorithm for the focused active inference problem; this algorithm is benchmarked in Section 6. A performance bound applicable to any focused selector is presented in Section 7.

## 2 Preliminaries

### 2.1 Markov Random Fields (MRFs)

Let $\mathcal{G} = (\mathcal{V}, \mathcal{E})$ be a Markov random field (MRF) with vertex set $\mathcal{V}$ and edge set $\mathcal{E}$. Let $u$ and $v$ be vertices of the graph $\mathcal{G}$. A $u$-$v$ *path* is a finite sequence of adjacent vertices, starting with vertex $u$ and terminating at vertex $v$, that does not repeat any vertex. Let $\mathcal{P}_{\mathcal{G}}(u, v)$ denote the set of all paths between distinct $u$ and $v$ in $\mathcal{G}$. If $|\mathcal{P}_{\mathcal{G}}(u, v)| > 0$, then $u$ and $v$ are *graph connected*. If $|\mathcal{P}_{\mathcal{G}}(u, v)| = 1$, then there is a *unique* path between $u$ and $v$, and denote the sole element of $\mathcal{P}_{\mathcal{G}}(u, v)$ by $P_{u:v}$.

If $|\mathcal{P}_{\mathcal{G}}(u, v)| = 1$ for all $u, v \in \mathcal{V}$, then $\mathcal{G}$ is a *tree*. If $|\mathcal{P}_{\mathcal{G}}(u, v)| \leq 1$ for all $u, v \in \mathcal{V}$, then $\mathcal{G}$ is a *forest*, i.e., a disjoint union of trees. A *chain* is a simple tree with diameter equal to the number of nodes. A chain is said to be embedded in graph $\mathcal{G}$ if the nodes in the chain comprise a unique path in $\mathcal{G}$.

For MRFs, the global Markov property relates connectivity in the graph to implied conditional independencies. If $D \subseteq \mathcal{V}$, then $\mathcal{G}_D = (D, \mathcal{E}_D)$ is the subgraph induced by $D$, with $\mathcal{E}_D = \mathcal{E} \cap (D \times D)$. For disjoint subsets $A, B, C \subset \mathcal{V}$, let $\mathcal{G}_{\backslash B}$ be the subgraph induced by $\mathcal{V} \setminus B$. The global Markov property holds that $\mathbf{x}_A \perp\!\!\!\perp \mathbf{x}_C \mid \mathbf{x}_B$ iff $|\mathcal{P}_{\mathcal{G}_{\backslash B}}(i, j)| = 0$ for all $i \in A$ and $j \in C$.

### 2.2 Gaussian Distributions in Information Form

Consider a random vector $\mathbf{x}$ distributed according to a multivariate Gaussian distribution $\mathcal{N}(\mu, \Lambda)$ with mean $\mu$ and (symmetric, positive definite) covariance $\Lambda > \mathbf{0}$. One could equivalently consider the information form $\mathbf{x} \sim \mathcal{N}^{-1}(\mathbf{h}, \mathbf{J})$ with precision matrix $\mathbf{J} = \Lambda^{-1} > \mathbf{0}$ and potential vector $\mathbf{h} = \mathbf{J}\mu$, for which $p_{\mathbf{x}}(\mathbf{x}) \propto \exp\{-\frac{1}{2}\mathbf{x}^T\mathbf{J}\mathbf{x} + \mathbf{h}^T\mathbf{x}\}$.

One can marginalize out or condition on a subset of random variables by considering a partition of $\mathbf{x}$ into two subvectors, $\mathbf{x}_1$ and $\mathbf{x}_2$, such that

$$\mathbf{x} = \begin{pmatrix} \mathbf{x}_1 \\ \mathbf{x}_2 \end{pmatrix} \sim \mathcal{N}^{-1}\left( \begin{pmatrix} \mathbf{h}_1 \\ \mathbf{h}_2 \end{pmatrix}, \begin{bmatrix} \mathbf{J}_{11} & \mathbf{J}_{12} \\ \mathbf{J}_{12}^T & \mathbf{J}_{22} \end{bmatrix} \right).$$

In the information form, the marginal distribution over $\mathbf{x}_1$ is $p_{\mathbf{x}_1}(\cdot) = \mathcal{N}^{-1}(\cdot; \mathbf{h}_1', \mathbf{J}_1')$, where $\mathbf{h}_1' = \mathbf{h}_1 - \mathbf{J}_{12}\mathbf{J}_{22}^{-1}\mathbf{h}_2$ and $\mathbf{J}_1' = \mathbf{J}_{11} - \mathbf{J}_{12}\mathbf{J}_{22}^{-1}\mathbf{J}_{12}^T$, the latter being the *Schur complement* of $\mathbf{J}_{22}$. Conditioning on a particular realization $\mathbf{x}_2$ of the random subvector $\mathbf{x}_2$ induces the conditional distribution $p_{\mathbf{x}_1|\mathbf{x}_2}(\mathbf{x}_1|\mathbf{x}_2) = \mathcal{N}^{-1}(\mathbf{x}_1; \mathbf{h}_{1|2}', \mathbf{J}_{11})$, where $\mathbf{h}_{1|2}' = \mathbf{h}_1 - \mathbf{J}_{12}\mathbf{x}_2$, and $\mathbf{J}_{11}$ is exactly the upper-left block submatrix of $\mathbf{J}$. (Note that the conditional precision matrix is *independent of the value* of the realized $\mathbf{x}_2$.)

If $\mathbf{x} \sim \mathcal{N}^{-1}(\mathbf{h}, \mathbf{J})$, where $\mathbf{h} \in \mathbb{R}^n$ and $\mathbf{J} \in \mathbb{R}^{n \times n}$, then the (differential) entropy of $\mathbf{x}$ is [9]

$$H(\mathbf{x}) = -\frac{1}{2} \log \left( (2\pi e)^n \cdot \det(\mathbf{J}) \right). \tag{1}$$

Likewise, for nonempty $A \subseteq \{1, \ldots, n\}$, and (possibly empty) $B \subseteq \{1, \ldots, n\} \setminus A$, let $\mathbf{J}'_{A|B}$ be the precision matrix parameterizing $p_{\mathbf{x}_A|\mathbf{x}_B}$. The conditional entropy of $\mathbf{x}_A \in \mathbb{R}^d$ given $\mathbf{x}_B$ is

$$H(\mathbf{x}_A|\mathbf{x}_B) = -\frac{1}{2} \log((2\pi e)^d \cdot \det(\mathbf{J}'_{A|B})). \tag{2}$$

The mutual information between $\mathbf{x}_A$ and $\mathbf{x}_B$ is

$$I(\mathbf{x}_A; \mathbf{x}_B) = H(\mathbf{x}_A) + H(\mathbf{x}_B) - H(\mathbf{x}_A, \mathbf{x}_B) = \frac{1}{2} \log \left( \frac{\det(\mathbf{J}'_{\{A,B\}})}{\det(\mathbf{J}'_A) \det(\mathbf{J}'_B)} \right), \tag{3}$$

which generally requires $\mathcal{O}(n^3)$ operations to compute via Schur complement.

## 2.3  Gaussian MRFs (GMRFs)

If $\mathbf{x} \sim \mathcal{N}^{-1}(\mathbf{h}, \mathbf{J})$, the conditional independence structure of $p_{\mathbf{x}}(\cdot)$ can be represented with a Gaussian MRF (GMRF) $\mathcal{G} = (\mathcal{V}, \mathcal{E})$, where $\mathcal{E}$ is determined by the sparsity pattern of $\mathbf{J}$ and the pairwise Markov property: $\{i, j\} \in \mathcal{E}$ iff $\mathbf{J}_{ij} \neq \mathbf{0}$.

In a scalar GMRF, $\mathcal{V}$ indexes scalar components of $\mathbf{x}$. In a *vectoral* GMRF, $\mathcal{V}$ indexes disjoint subvectors of $\mathbf{x}$, each of potentially different dimension. The block submatrix $\mathbf{J}_{ii}$ can be thought of as specifying the sparsity pattern of the scalar micro-network within the vectoral macro-node $i \in \mathcal{V}$.

## 2.4  Gaussian Belief Propagation (GaBP)

If $\mathbf{x}$ can be partitioned into $n$ subvectors of dimension at most $d$, and the resulting graph is tree-shaped, then all marginal precision matrices $\mathbf{J}'_i$, $i \in \mathcal{V}$ can be computed by Gaussian belief propagation (GaBP) [10] in $\mathcal{O}(n \cdot d^3)$. For such trees, one can also compute all edge marginal precision matrices $\mathbf{J}'_{\{i,j\}}$, $\{i, j\} \in \mathcal{E}$, with the same asymptotic complexity of $\mathcal{O}(n \cdot d^3)$.

In light of (3), pairwise MI quantities *between adjacent nodes* $i$ and $j$ may be expressed as

$$\begin{aligned} I(\mathbf{x}_i; \mathbf{x}_j) &= H(\mathbf{x}_i) + H(\mathbf{x}_j) - H(\mathbf{x}_i, \mathbf{x}_j), \\ &= -\frac{1}{2} \ln \det(\mathbf{J}'_i) - \frac{1}{2} \ln \det(\mathbf{J}'_j) + \frac{1}{2} \ln \det(\mathbf{J}'_{\{i,j\}}), \quad \{i, j\} \in \mathcal{E}, \end{aligned} \tag{4}$$

i.e., purely in terms of node and edge marginal precision matrices. Thus, GaBP provides a way of computing all *local* pairwise MI quantities in $\mathcal{O}(n \cdot d^3)$.

Note that Gaussian trees comprise an important class of distributions that subsumes Gaussian hidden Markov models (HMMs), and GaBP on trees is a generalization of the Kalman filtering/smoothing algorithms that operate on HMMs. Moreover, the graphical inference community appears to best understand the convergence of message passing algorithms for continuous distributions on subclasses of multivariate Gaussians (e.g., tree-shaped [10], walk-summable [11], and feedback-separable [12] models, among others).

# 3  Problem Statement

Let $p_{\mathbf{x}}(\cdot) = \mathcal{N}^{-1}(\cdot; \mathbf{h}, \mathbf{J})$ be represented by GMRF $\mathcal{G} = (\mathcal{V}, \mathcal{E})$, and consider a partition of $\mathcal{V}$ into the subsets of latent nodes $\mathcal{U}$ and observable nodes $\mathcal{S}$, with $\mathcal{R} \subseteq \mathcal{U}$ denoting the subset of *relevant* latent variables (i.e., those to be inferred). Given a cost function $c : 2^{\mathcal{S}} \to \mathbb{R}_{\geq 0}$ over subsets of observations, and a budget $\beta \in \mathbb{R}_{\geq 0}$, the *focused* active inference problem is

$$\begin{aligned} \text{maximize}_{\mathcal{A} \subseteq \mathcal{S}} \quad & I(\mathbf{x}_{\mathcal{R}}; \mathbf{x}_{\mathcal{A}}) \\ \text{s.t.} \quad & c(\mathcal{A}) \leq \beta. \end{aligned} \tag{5}$$

The focused active inference problem in (5) is distinguished from the *unfocused* active inference problem

$$\begin{aligned} \text{maximize}_{\mathcal{A} \subseteq \mathcal{S}} \quad & I(\mathbf{x}_{\mathcal{U}}; \mathbf{x}_{\mathcal{A}}) \\ \text{s.t.} \quad & c(\mathcal{A}) \leq \beta, \end{aligned} \tag{6}$$

which considers the entirety of the latent state $\mathcal{U} \supseteq \mathcal{R}$ to be of interest. Both problems are known to be **NP**-hard [13, 14].

By the chain rule and nonnegativity of MI, $I(\mathbf{x}_{\mathcal{U}}; \mathbf{x}_{\mathcal{A}}) = I(\mathbf{x}_{\mathcal{R}}; \mathbf{x}_{\mathcal{A}}) + I(\mathbf{x}_{\mathcal{U} \setminus \mathcal{R}}; \mathbf{x}_{\mathcal{A}} \mid \mathbf{x}_{\mathcal{R}}) \geq I(\mathbf{x}_{\mathcal{R}}; \mathbf{x}_{\mathcal{A}})$, for any $\mathcal{A} \subseteq \mathcal{S}$. Therefore, maximizing unfocused MI does not imply maximizing focused MI. Focused active inference must be posed as a separate problem to avoid the situation where the observation selector becomes fixated on inferring nuisance variables as a result of $I(\mathbf{x}_{\mathcal{U} \setminus \mathcal{R}}; \mathbf{x}_{\mathcal{A}} \mid \mathbf{x}_{\mathcal{R}})$ being included implicitly in the valuation. In fact, an unfocused selector can perform arbitrarily poorly with respect to a focused metric, as the following example illustrates.

*Example* 1. Consider a scalar GMRF over a four-node chain (Figure 1a), whereby $J_{13} = J_{14} = J_{24} = 0$ by the pairwise Markov property, with $\mathcal{R} = \{2\}$, $\mathcal{S} = \{1, 4\}$, $c(\mathcal{A}) = |\mathcal{A}|$ (i.e., unit-cost observations), and $\beta = 1$. The optimal unfocused decision rule $\mathcal{A}^*_{(UF)} = \text{argmax}_{a \in \{1,4\}} I(\mathsf{x}_2, \mathsf{x}_3; \mathsf{x}_a)$ can be shown, by conditional independence and positive definiteness of $\mathbf{J}$, to reduce to

$$|J_{34}| \underset{\mathcal{A}^*_{(UF)}=\{1\}}{\overset{\mathcal{A}^*_{(UF)}=\{4\}}{\gtrless}} |J_{12}|,$$

independent of $J_{23}$, which parameterizes the edge potential between nodes 2 and 3. Conversely, the optimal focused decision rule $\mathcal{A}^*_{(F)} = \text{argmax}_{a \in \{1,4\}} I(\mathsf{x}_2; \mathsf{x}_a)$ can be shown to be

$$|J_{23}| \cdot \mathbf{1}_{\left\{ J_{34}^2 - J_{12}^2 J_{34}^2 - J_{12}^2 \geq 0 \right\}} \underset{\mathcal{A}^*_{(F)}=\{1\}}{\overset{\mathcal{A}^*_{(F)}=\{4\}}{\gtrless}} \sqrt{\frac{(1 - J_{34}^2)J_{12}^2}{J_{34}^2}},$$

where $\mathbf{1}_{\{\cdot\}}$ is the indicator function, which evaluates to 1 when its argument is true and 0 otherwise. The loss associated with optimizing the "wrong" information measure is demonstrated in Figure 1b. The reason for this loss is that as $|J_{23}| \to 0^+$, the information that node 3 can convey about node 2 also approaches zero, although the unfocused decision rule is oblivious to this fact.

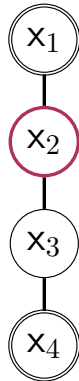

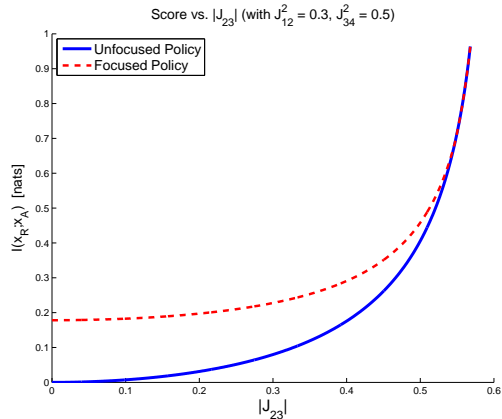

(a) Graphical model.          (b) Policy comparison.

Figure 1: (a) Graphical model for the four-node chain example. (b) Unfocused vs. focused policy comparison. There exists a range of values for $|J_{23}|$ such that the unfocused and focused policies coincide; however, as $|J_{23}| \to 0^+$, the unfocused policy approaches complete performance loss with respect to the focused measure.

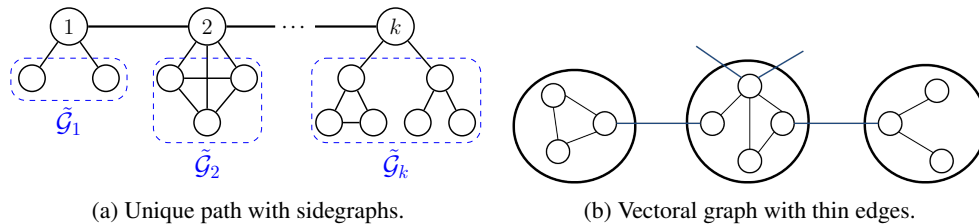

(a) Unique path with sidegraphs.  (b) Vectoral graph with thin edges.

Figure 2: (a) Example of a nontree graph $\mathcal{G}$ with a unique path $\bar{P}_{1:k}$ between nodes 1 and $k$. The "sidegraph" attached to each node $i \in \bar{P}_{1:k}$ is labeled as $\tilde{\mathcal{G}}_i$. (b) Example of a vectoral graph with thin edges, with internal (scalar) structure depicted.

## 4 Nonlocal MI Decomposition

For GMRFs with $n$ nodes indexing $d$-dimensional random subvectors, $I(\mathbf{x}_\mathcal{R}; \mathbf{x}_\mathcal{A})$ can be computed exactly in $\mathcal{O}((nd)^3)$ via Schur complements/inversions on the precision matrix $\mathbf{J}$. However, certain graph structures permit the computation via belief propagation of all local pairwise MI terms $I(\mathbf{x}_i; \mathbf{x}_j)$, for adjacent nodes $i, j \in \mathcal{V}$ in $\mathcal{O}(n \cdot d^3)$ – a substantial savings for large networks. This section describes a transformation of nonlocal MI between uniquely path-connected nodes that permits a decomposition into the sum of transformed local MI quantities, i.e., those relating adjacent nodes in the graph. Furthermore, the local MI terms can be transformed in constant time, yielding an $\mathcal{O}(n \cdot d^3)$ for computing any pairwise nonlocal MI quantity coinciding with a unique path.

**Definition 1** (Warped MI). For disjoint subsets $A, B, C \subseteq \mathcal{V}$, the warped mutual information measure $W : 2^\mathcal{V} \times 2^\mathcal{V} \times 2^\mathcal{V} \to (-\infty, 0]$ is defined such that $W(A; B|C) \triangleq \frac{1}{2} \log \left(1 - \exp\{-2I(\mathbf{x}_A; \mathbf{x}_B | \mathbf{x}_C)\}\right)$.

For convenience, let $W(i; j|C) \triangleq W(\{i\}; \{j\}|C)$ for $i, j \in \mathcal{V}$.

*Remark* 2. For $i, j \in \mathcal{V}$ indexing scalar nodes, the warped MI of Definition 1 reduces to $W(i; j) = \log |\rho_{ij}|$, where $\rho_{ij} \in [-1, 1]$ is the correlation coefficient between scalar r.v.s $\mathbf{x}_i$ and $\mathbf{x}_j$. The measure $\log |\rho_{ij}|$ has long been known to the graphical model learning community as an "additive tree distance" [15, 16], and our decomposition for vectoral graphs is a novel application for sensor selection problems. To the best of the authors' knowledge, the only other distribution class with established additive distances are tree-shaped symmetric discrete distributions [16], which require a very limiting parameterization of the potentials functions defined over edges in the factorization of the joint distribution.

**Proposition 3** (Scalar Nonlocal MI Decomposition). *For any GMRF $\mathcal{G} = (\mathcal{V}, \mathcal{E})$ where $\mathcal{V}$ indexes scalar random variables, if $|\mathcal{P}_\mathcal{G}(u, v)| = 1$ for distinct vertices $u, v \in \mathcal{V}$, then for any $C \subseteq \mathcal{V} \setminus \{u, v\}$, $I(\mathbf{x}_u; \mathbf{x}_v | \mathbf{x}_C)$ can be decomposed as*

$$W(u; v|C) = \sum_{\{i,j\} \in \bar{\mathcal{E}}_{u:v}} W(i; j|C), \tag{7}$$

*where $\bar{\mathcal{E}}_{u:v}$ is the set of edges joining consecutive nodes of $\bar{P}_{u:v}$, the unique path between $u$ and $v$ and sole element of $\mathcal{P}_\mathcal{G}(u, v)$.*

(Proofs of this and subsequent propositions can be found in the supplementary material.)

*Remark* 4. Proposition 3 requires only that the path between vertices $u$ and $v$ be unique. If $\mathcal{G}$ is a tree, this is obviously satisfied. However, the result holds on any graph for which: the subgraph induced by $\bar{P}_{u:v}$ is a chain; and every $i \in \bar{P}_{u:v}$ separates $N(i) \setminus \bar{P}_{u:v}$ from $\bar{P}_{u:v} \setminus \{i\}$, where $N(i) \triangleq \{j : \{i, j\} \in \mathcal{E}\}$ is the neighbor set of $i$. See Figure 2a for an example of a nontree graph with a unique path.

**Definition 5** (Thin Edges). An edge $\{i, j\} \in \mathcal{E}$ of GMRF $\mathcal{G} = (\mathcal{V}, \mathcal{E}; \mathbf{J})$ is thin if the corresponding submatrix $\mathbf{J}_{ij}$ has exactly one nonzero scalar component. (See Figure 2b.)

For vectoral problems, each node may contain a subnetwork of arbitrarily connected scalar random variables (see Figure 2b). Under the assumption of thin edges (Definition 5), a unique path between nodes $u$ and $v$ must enter interstitial nodes through one scalar r.v. and leave through one scalar

r.v. Therefore, let $\zeta_i(u,v|C) \in (-\infty, 0]$ denote the warped MI between the enter and exit r.v.s of interstitial vectoral node $i$ on $\bar{P}_{u:v}$, with conditioning set $C \subseteq \mathcal{V} \setminus \{u,v\}$.[1] Note that $\zeta_i(u,v|C)$ can be computed online in $\mathcal{O}(d^3)$ via local marginalization given $\mathbf{J}'_{i|C}$, which is an output of GaBP.

**Proposition 6** (Vectoral Nonlocal MI Decomposition). *For any GMRF $\mathcal{G} = (\mathcal{V}, \mathcal{E})$ where $\mathcal{V}$ indexes random vectors of dimension at most $d$ and the edges in $\mathcal{E}$ are thin, if $|\mathcal{P}_{\mathcal{G}}(u,v)| = 1$ for distinct vertices $u, v \in \mathcal{V}$, then for any $C \subseteq \mathcal{V} \setminus \{u,v\}$, $I(\mathbf{x}_u; \mathbf{x}_v | \mathbf{x}_C)$ can be decomposed as*

$$W(u;v|C) = \sum_{\{i,j\} \in \bar{\mathcal{E}}_{u:v}} W(i;j|C) + \sum_{i \in \bar{P}_{u:v} \setminus \{u,v\}} \zeta_i(u,v|C). \tag{8}$$

## 5  (Distributed) Focused Greedy Selection

The nonlocal MI decompositions of Section 4 can be used to efficiently solve the focused greedy selection problem, which at each iteration, given the subset $\mathcal{A} \subset \mathcal{S}$ of previously selected observable random variables, is

$$\operatorname*{argmax}_{\{y \in \mathcal{S} \setminus \mathcal{A} \,:\, c(y) \leq \beta - c(\mathcal{A})\}} I(\mathbf{x}_{\mathcal{R}}; \mathbf{x}_y \mid \mathbf{x}_{\mathcal{A}}).$$

To proceed, first consider the singleton case $\mathcal{R} = \{r\}$ for $r \in \mathcal{U}$. Running GaBP on the graph $\mathcal{G}$ conditioned on $\mathcal{A}$ and subsequently computing all terms $W(i;j|\mathcal{A}), \forall \{i,j\} \in \mathcal{E}$ incurs a computational cost of $\mathcal{O}(n \cdot d^3)$. Once GaBP has converged, node $r$ authors an "r-message" with the value 0. Each neighbor $i \in N(r)$ receives that message with value modified by $W(r;i|\mathcal{A})$; there is no $\zeta$ term because there are no interstitial nodes between $r$ and its neighbors. Subsequently, each $i \in N(r)$ messages its neighbors $j \in N(i) \setminus \{r\}$, modifying the value of its r-message by $W(i;j|\mathcal{A}) + \zeta_i(r,j|\mathcal{A})$, the latter term being computed online in $\mathcal{O}(d^3)$ from $\mathbf{J}'_{i|\mathcal{A}}$, itself an output of GaBP.[2] Then $j$ messages $N(j) \setminus \{i\}$, and so on down to the leaves of the tree. Since there are at most $n-1$ edges in a forest, the total cost of dissemination is still $\mathcal{O}(n \cdot d^3)$, after which all nodes $y$ in the same component as $r$ will have received an r-message whose value on arrival is $W(r;y|\mathcal{A})$, from which $I(\mathbf{x}_r; \mathbf{x}_y | \mathcal{A})$ can be computed in constant time. Thus, for $|\mathcal{R}| = 1$, *all* scores $I(\mathbf{x}_{\mathcal{R}}; \mathbf{x}_y | \mathbf{x}_{\mathcal{A}})$ for $y \in \mathcal{S} \setminus \mathcal{A}$ can collectively be computed at each iteration of the greedy algorithm in $\mathcal{O}(n \cdot d^3)$.

Now consider $|\mathcal{R}| > 1$. Let $R = (r_1, \ldots, r_{|\mathcal{R}|})$ be an ordering of the elements of $\mathcal{R}$, and let $R_k$ be the first $k$ elements of $R$. Then, by the chain rule of mutual information, $I(\mathbf{x}_{\mathcal{R}}; \mathbf{x}_y \mid \mathbf{x}_{\mathcal{A}}) = \sum_{k=1}^{|\mathcal{R}|} I(\mathbf{x}_{r_k}; \mathbf{x}_y \mid \mathbf{x}_{\mathcal{A} \cup R_{k-1}})$, $y \in \mathcal{S} \setminus \mathcal{A}$, where each term in the sum is a pairwise (potentially nonlocal) MI evaluation. The implication is that one can run $|\mathcal{R}|$ separate instances of GaBP, each using a different conditioning set $\mathcal{A} \cup R_{k-1}$, to compute "node and edge weights" ($W$ and $\zeta$ terms) for the r-message passing scheme outlined above. The chain rule suggests one should then sum the unwarped r-scores of these $|\mathcal{R}|$ instances to yield the scores $I(\mathbf{x}_{\mathcal{R}}; \mathbf{x}_y | \mathbf{x}_{\mathcal{A}})$ for $y \in \mathcal{S} \setminus \mathcal{A}$. The total cost of a greedy update is then $\mathcal{O}\left(|\mathcal{R}| \cdot nd^3\right)$.

One of the benefits of the focused greedy selection algorithm is its amenability to parallelization. All quantities needed to form the $W$ and $\zeta$ terms are derived from GaBP, which is parallelizable and guaranteed to converge on trees in at most $\operatorname{diam}(\mathcal{G})$ iterations [10]. Parallelization reallocates the expense of quantification across networked computational resources, often leading to faster solution times and enabling larger problem instantiations than are otherwise permissible. However, full parallelization, wherein each node $i \in \mathcal{V}$ is viewed as separate computing resource, incurs a multiplicative overhead of $\mathcal{O}(\operatorname{diam}(\mathcal{G}))$ due to each $i$ having to send $|N(i)|$ messages $\operatorname{diam}(\mathcal{G})$ times, yielding local communication costs of $\mathcal{O}(\operatorname{diam}(\mathcal{G})|N(i)| \cdot d^3)$ and overall complexity of $\mathcal{O}(\operatorname{diam}(\mathcal{G}) \cdot |\mathcal{R}| \cdot nd^3)$. This overhead can be alleviated by instead assigning to every computational resource a connected subgraph of $\mathcal{G}$.

It should also be noted that if the quantification is instead performed using serial BP – which can be conceptualized as choosing an arbitrary root, collecting messages from the leaves up to the root, and disseminating messages back down again – a factor of 2 savings can be achieved for $R_2, \ldots, R_{|\mathcal{R}|}$ by noting that in moving between instances $k$ and $k+1$, only $r_k$ is added to the conditioning set. Therefore, by reassigning $r_k$ as the root for the BP instance associated with $r_{k+1}$ (i.e., $\mathcal{A} \cup R_k$ as the conditioning set), only the second half of the message passing schedule (disseminating messages from the root to the leaves) is necessary. We subsequently refer to this trick as "caching."

# 6 Experiments

To benchmark the runtime performance of the algorithm in Section 5, we implemented its serial GaBP variant in Java, with and without the caching trick described above.

We compare our algorithm with greedy selectors that use matrix inversion (with cubic complexity) to compute nonlocal mutual information measures. Let $\mathcal{S}_{\text{feas}} \coloneqq \{y \in \mathcal{S} \setminus \mathcal{A} : c(y) \le \beta - c(\mathcal{A})\}$. At each iteration of the greedy selector, the blocked inversion-based quantifier computes first $\mathbf{J}'_{\mathcal{R} \cup \mathcal{S}_{\text{feas}} | \mathcal{A}}$ (entailing a block marginalization of nuisances), from which $\mathbf{J}'_{\mathcal{R} | \mathcal{A}}$ and $\mathbf{J}'_{\mathcal{R} | \mathcal{A} \cup y}, \forall y \in \mathcal{S}_{\text{feas}}$, are computed. Then $I(\mathbf{x}_{\mathcal{R}}; \mathbf{x}_y \mid \mathbf{x}_{\mathcal{A}}), \forall y \in \mathcal{S}_{\text{feas}}$, are computed via a variant of (3). The naïve inversion-based quantifier computes $I(\mathbf{x}_{\mathcal{R}}; \mathbf{x}_y \mid \mathbf{x}_{\mathcal{A}}), \forall y \in \mathcal{S}_{\text{feas}}$, "from scratch" by using separate Schur complements of $\mathbf{J}$ submatrices and not storing intermediate results. The inversion-based quantifiers were implemented in Java using the Colt sparse matrix libraries [17].

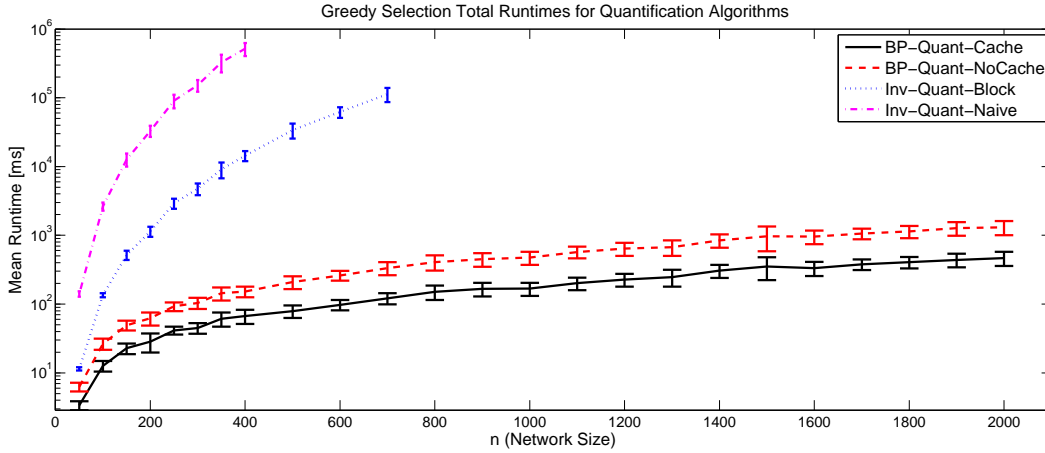

Figure 3: Performance of GaBP-based and inversion-based quantifiers used in greedy selectors. For each $n$, the mean of the runtimes over 20 random scalar problem instances is displayed. Our `BP-Quant` algorithm of Section 5 empirically has approximately linear complexity; caching reduces the mean runtime by a factor of approximately 2.

Figure 3 shows the comparative mean runtime performance of each of the quantifiers for scalar networks of size $n$, where the mean is taken over the 20 problem instances proposed for each value of $n$. Each problem instance consists of a randomly generated, symmetric, positive-definite, tree-shaped precision matrix $\mathbf{J}$, along with a randomly labeled $\mathcal{S}$ (such that, arbitrarily, $|\mathcal{S}| = 0.3|\mathcal{V}|$) and $\mathcal{R}$ (such that $|\mathcal{R}| = 5$), as well as randomly selected budget and heterogeneous costs defined over $\mathcal{S}$. Note that all selectors return the same greedy selection; we are concerned with how the decompositions proposed in this paper aid in the computational performance. In the figure, it is clear that the GaBP-based quantification algorithms of Section 5 vastly outperform both inversion-based methods; for relatively small $n$, the solution times for the inversion-based methods became prohibitively long. Conversely, the behavior of the BP-based quantifiers empirically confirms the asymptotic $\mathcal{O}(n)$ complexity of our method for scalar networks.

# 7 Performance Bounds

Due to the presence of nuisances in the model, even if the subgraph induced by $\mathcal{S}$ is completely disconnected, it is not always the case that the nodes in $\mathcal{S}$ are conditionally independent when conditioned on only the *relevant* latent set $\mathcal{R}$. Lack of conditional independence means one cannot guarantee submodularity of the information measure, as per [6]. Our approach will be to augment $\mathcal{R}$ such that submodularity is guaranteed and relate the performance bound to this augmented set.

Let $\hat{\mathcal{R}}$ be any subset such that $\mathcal{R} \subset \hat{\mathcal{R}} \subseteq \mathcal{U}$ and such that nodes in $\mathcal{S}$ are conditionally independent conditioned on $\hat{\mathcal{R}}$. Then, by Corollary 4 of [6], $I(\mathbf{x}_{\hat{\mathcal{R}}}; \mathbf{x}_{\mathcal{A}})$ is submodular and nondecreasing on $\mathcal{S}$. Additionally, for the case of unit-cost observations (i.e., $c(\mathcal{A}) = |\mathcal{A}|$ for all $\mathcal{A} \subseteq \mathcal{S}$), a greedily selected subset $\mathcal{A}_\beta^g(\hat{\mathcal{R}})$ of cardinality $\beta$ satisfies the performance bound

$$I(\hat{\mathcal{R}}; \mathcal{A}_\beta^g(\hat{\mathcal{R}})) \geq \left(1 - \frac{1}{e}\right) \max_{\{\mathcal{A} \subseteq \mathcal{S}: |\mathcal{A}| \leq \beta\}} I(\hat{\mathcal{R}}; \mathcal{A}) \tag{9}$$

$$= \left(1 - \frac{1}{e}\right) \max_{\{\mathcal{A} \subseteq \mathcal{S}: |\mathcal{A}| \leq \beta\}} [I(\mathcal{R}; \mathcal{A}) + I(\hat{\mathcal{R}} \setminus \mathcal{R}; \mathcal{A}|\mathcal{R})] \tag{10}$$

$$\geq \left(1 - \frac{1}{e}\right) \max_{\{\mathcal{A} \subseteq \mathcal{S}: |\mathcal{A}| \leq \beta\}} I(\mathcal{R}; \mathcal{A}), \tag{11}$$

where (9) is due to [6], (10) to the chain rule of MI, and (11) to the nonnegativity of MI. The following proposition follows immediately from (11).

**Proposition 7.** *For any set $\hat{\mathcal{R}}$ such that $\mathcal{R} \subset \hat{\mathcal{R}} \subseteq \mathcal{U}$ and nodes in $\mathcal{S}$ are conditionally independent conditioned on $\hat{\mathcal{R}}$, provided $I(\hat{\mathcal{R}}; \mathcal{A}_\beta^g(\hat{\mathcal{R}})) > 0$, an online-computable performance bound for any $\bar{\mathcal{A}} \subseteq \mathcal{S}$ in the original focused problem with relevant set $\mathcal{R}$ and unit-cost observations is*

$$I(\mathcal{R}; \bar{\mathcal{A}}) \geq \underbrace{\left[\frac{I(\mathcal{R}; \bar{\mathcal{A}})}{I(\hat{\mathcal{R}}; \mathcal{A}_\beta^g(\hat{\mathcal{R}}))}\right] \left(1 - \frac{1}{e}\right)}_{\triangleq \, \delta_{\mathcal{R}}(\bar{\mathcal{A}}, \hat{\mathcal{R}})} \max_{\{\mathcal{A} \subseteq \mathcal{S}: |\mathcal{A}| \leq \beta\}} I(\mathcal{R}; \mathcal{A}). \tag{12}$$

Proposition 7 can be used at runtime to determine what percentage $\delta_{\mathcal{R}}(\bar{\mathcal{A}}, \hat{\mathcal{R}})$ of the optimal objective is guaranteed, for any focused selector, despite the lack of conditional independence of $\mathcal{S}$ conditioned on $\mathcal{R}$. In order to compute the bound, a greedy heuristic running on a separate, surrogate problem with $\hat{\mathcal{R}}$ as the relevant set is required. Finding an $\hat{\mathcal{R}} \supset \mathcal{R}$ providing the tightest bound is an area of future research.

# 8 Conclusion

In this paper, we have considered the sensor selection problem on multivariate Gaussian distributions that, in order to preserve a parsimonious representation, contain nuisances. For pairs of nodes connected in the graph by a unique path, there exist decompositions of nonlocal mutual information into local MI measures that can be computed efficiently from the output of message passing algorithms. For tree-shaped models, we have presented a greedy selector where the computational expense of quantification can be distributed across nodes in the network. Despite deficiency in conditional independence of observations, we have derived an online-computable performance bound based on an augmentation of the relevant set. Future work will consider extensions of the MI decomposition to graphs with nonunique paths and/or non-Gaussian distributions, as well as extend the analysis of augmented relevant sets to derive tighter performance bounds.

# Acknowledgments

The authors thank John W. Fisher III, Myung Jin Choi, and Matthew Johnson for helpful discussions during the preparation of this paper. This work was supported by DARPA Mathematics of Sensing, Exploitation and Execution (MSEE).

## Footnotes

[1]As node $i$ may have additional neighbors that are not on the $u$-$v$ path, using the notation $\zeta_i(u,v|C)$ is a convenient way to implicitly specify the enter/exit scalar r.v.s associated with the path. Any unique path subsuming $u$-$v$, or any unique path subsumed in $u$-$v$ for which $i$ is interstitial, will have equivalent $\zeta_i$ terms.

[2]If $i$ is in the conditioning set, its outgoing message can be set to be $-\infty$, so that the nodes it blocks from reaching $r$ see an apparent information score of 0. Alternatively, $i$ could simply choose not to transmit r-messages to its neighbors.

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
