[Supplementary Material]

# Sensor Selection in High-Dimensional Gaussian Trees with Nuisances – Supplementary Material

**Daniel Levine**
MIT LIDS
dlevine@mit.edu

**Jonathan P. How**
MIT LIDS
jhow@mit.edu

## Proofs

In order to prove Proposition 3, it is convenient to first prove the following lemma.

**Lemma A.** *Consider a connected GMRF $\mathcal{G} = (\mathcal{V}, \mathcal{E}; \mathbf{J})$ parameterized by precision matrix $\mathbf{J}$ and a unique path $\bar{P}$ embedded in $\mathcal{G}$. The marginal precision matrix $\mathbf{J}'_{\bar{P}}$ has block off-diagonal elements identical to those in the submatrix of $\mathbf{J}$ corresponding to variables in $\bar{P}$, and block diagonal elements that are the Schur complements of the submatrices corresponding to the sidegraphs separated by $\bar{P}$.*

*Proof.* Assume without loss of generality that the unique path under consideration is $(1, 2, \ldots, k - 1, k)$. Because $\bar{P}_{1:k}$ is unique, the graph $\tilde{\mathcal{G}}$ induced by $\mathcal{V} \setminus \bar{P}_{1:k}$ can be thought of as the union of conditionally independent "sidegraphs" $\tilde{\mathcal{G}}_1, \ldots, \tilde{\mathcal{G}}_k$, each of which is connected in $\mathcal{G}$ to a single node in $\bar{P}_{1:k}$ (see Figure 2a). Let $\mathbf{J}_{\bar{P}_{1:k}}$ denote the (block tridiagonal) matrix parameterizing the joint potential (i.e., the product of singleton and edge potentials in the factorization of the full joint distribution of $\mathbf{x}$) over the chain $(1, \ldots, k)$. For all $i \in \{1, \ldots, k\}$, let $\mathbf{J}_{i, \tilde{\mathcal{G}}_i}$ be the matrix parameterizing the potentials over edges between $i$ and $N(i) \setminus \bar{P}_{1:k}$. Likewise, let $\mathbf{J}_{\tilde{\mathcal{G}}_i}$ denote the matrix parameterizing the joint potential over the subgraph $\tilde{\mathcal{G}}_i$.

Now consider a permutation to the last $n - k$ components of $\mathbf{J}$ such that $\mathbf{J}_{\tilde{\mathcal{G}}_1}, \ldots, \mathbf{J}_{\tilde{\mathcal{G}}_k}$ are ordered as such, whereby

$$\mathbf{J} \triangleq \left[ \begin{array}{c|c} \mathbf{J}_{\bar{P}_{1:k}} & \mathbf{J}_{\bar{P}_{1:k}, \tilde{\mathcal{G}}} \\ \hline \mathbf{J}^T_{\bar{P}_{1:k}, \tilde{\mathcal{G}}} & \mathbf{J}_{\tilde{\mathcal{G}}} \end{array} \right].$$

In this permuted matrix, $\mathbf{J}_{\tilde{\mathcal{G}}}$ is block diagonal – due to conditional independence of the sidegraphs – with elements $\mathbf{J}_{\tilde{\mathcal{G}}_i}$. Similarly, the upper-right block submatrix $\mathbf{J}_{\bar{P}_{1:k}, \tilde{\mathcal{G}}}$ is also block diagonal with elements $\mathbf{J}_{i, \tilde{\mathcal{G}}_i}$. Thus, the marginal distribution $p_{\mathbf{x}_1, \ldots, \mathbf{x}_k}$ is parameterized by a precision matrix

$$\mathbf{J}'_{\bar{P}_{1:k}} = \mathbf{J}_{\bar{P}_{1:k}} - \mathbf{J}_{\bar{P}_{1:k}, \tilde{\mathcal{G}}} \mathbf{J}^{-1}_{\tilde{\mathcal{G}}} \mathbf{J}^T_{\bar{P}_{1:k}, \tilde{\mathcal{G}}},$$

where the subtractive term is a product of block diagonal matrices and, thus, is itself a block diagonal matrix. Therefore, the marginal precision matrix $\mathbf{J}'_{\bar{P}_{1:k}}$ has block off-diagonal elements identical to those of the submatrix $\mathbf{J}_{\bar{P}_{1:k}}$ of the (full) joint precision matrix; each block diagonal element is the Schur complement of each $\mathbf{J}_{\tilde{\mathcal{G}}_i}$, $i = 1, \ldots, k$. $\qquad \square$

*Remark* B. Lemma A implies that if Proposition 3 holds for any chain of length $k$ between nodes $u$ and $v$, it must also hold for the more general class of graphs in which $|\mathcal{P}(u, v)| = 1$ (i.e., there is a unique path between $u$ and $v$, but there are sidegraphs attached to each vertex in the path). Therefore, we need only prove Proposition 3 for chains of arbitrary length. Furthermore, conditioning only severs nodes from the graph component considered; provided $C$ is not a separator for $u, v$, in which case $I(\mathbf{x}_u; \mathbf{x}_v | \mathbf{x}_C) = 0$, we need only prove Proposition 3 for the case where $C = \emptyset$.

*Proof of Proposition 3.* We proceed by induction on the length $k$ of the chain. The base case considered is a chain of length $k = 3$, for which the precision matrix is

$$\begin{bmatrix} J_{11} & J_{12} & 0 \\ J_{12} & J_{22} & J_{23} \\ 0 & J_{23} & J_{33} \end{bmatrix}.$$

By Remark 2, we need only show that $|\rho_{13}| = |\rho_{12}| \cdot |\rho_{23}|$. We have

$$\mathbf{J}^{-1} = \frac{1}{\det(\mathbf{J})} \begin{bmatrix} J_{22}J_{33} - J_{23}^2 & -J_{12}J_{33} & J_{12}J_{23} \\ -J_{12}J_{33} & J_{11}J_{33} & -J_{11}J_{23} \\ J_{12}J_{23} & -J_{11}J_{23} & J_{11}J_{22} - J_{12}^2 \end{bmatrix},$$

from which it is straightforward to confirm that

$$\rho_{12} = -J_{12}J_{33}/\sqrt{J_{11}J_{33}(J_{22}J_{33} - J_{23}^2)}$$

$$\rho_{23} = -J_{11}J_{23}/\sqrt{J_{11}J_{33}(J_{11}J_{22} - J_{12}^2)}$$

$$\rho_{13} = J_{12}J_{23}/\sqrt{(J_{11}J_{22} - J_{12}^2)(J_{22}J_{33} - J_{23}^2)}$$

$$= \rho_{12} \cdot \rho_{23},$$

thus proving the base case.

Now, assume the result of the proposition holds for a unique path $\bar{P}_{1:k}$ of length $k$ embedded in graph $\mathcal{G}$, and consider node $k + 1 \in N(k) \setminus \bar{P}_{1:k}$. By Lemma A, we can restrict our attention to the marginal chain over $(1, \ldots, k, k+1)$. Pairwise decomposition on subchains of length $k$ yields an expression for $I(\mathsf{x}_1; \mathsf{x}_k)$, which by (4) can alternatively be expressed in terms of the determinants of marginal precision matrices. Therefore, if one marginalizes nodes in $\{2, \ldots, k-1\}$ (under any elimination ordering), one is left with a graph over nodes $1$, $k$, and $k + 1$. The MI between $k$ and $k + 1$, which are adjacent in $\mathcal{G}$, can be computed from the local GaBP messages comprising the marginal node and edge precision matrices. On this 3-node network, for which the base case holds,

$$W(1; k+1) = W(k; k+1) + W(1; k)$$
$$= W(k; k+1) + \sum_{\{i,j\} \in \bar{\mathcal{E}}_{1:k}} W(i; j)$$
$$= \sum_{\{i,j\} \in \bar{\mathcal{E}}_{1:k+1}} W(i; j).$$

Therefore, Proposition 3 holds for chains of arbitrary length and, by Lemma A, unique paths of arbitrary length in GMRFs with scalar variables. $\quad\square$

*Proof of Proposition 6.* The proof follows closely that of Proposition 3. By Lemma A, assume without loss of generality that the unique path under consideration is a vectoral chain of length $k$ with sequentially indexed nodes, i.e., $(1, 2, \ldots, k-1, k)$. Thinness of edges $\{i, j\} \in \mathcal{E}$ implies $W(i; j) = \log|\rho_{ij}|$, as before. Let $i \in \{2, \ldots, k-1\}$ be an arbitrary interstitial node. On section $(i-1, i, i+1)$ of the chain, thinness of $\{i-1, i\}$ and $\{i, i+1\}$ implies that on $\bar{P}_{1:k}$, there exists one inlet to and one outlet from $i$. Let $m$ and $q$ denote the column of $\mathbf{J}_{i-1,i}$ and row of $\mathbf{J}_{i,i+1}$, respectively, containing nonzero elements. Then the path through the internal structure of node $i$ can be simplified by marginalizing out, as computed via Schur complement from $\mathbf{J}'_i$ in $\mathcal{O}(d^3)$, all scalar elements of $i$ except $m$ and $q$. Thus, $\zeta_i(u, v)$ is merely the warped mutual information between $m$ and $q$, and problem reduces to a scalar chain with alternating $W$ and $\zeta$ terms. $\quad\square$