[Reviews · NeurIPS 2013]

Submitted by Assigned_Reviewer_1

The paper studies the problem of sensor selection on multivariate Gaussian distributions, where only a subset of latent variables is of inferential interest. The paper shows that there exist decompositions of nonlocal mutual information into local information measures which can be computed efficiently. These ideas lead to a new greedy selector. Experimental results demonstrate the effectiveness of this method.

The paper is well written, and the main idea is clear. I would recommend for accepting, although I have to mention that I have no background in Markov random fields, so I can not decide whether the contribution here is significant.
Summary: The paper is well written, and the main idea is clear. I would recommend for accepting, although I have to mention that I have no background in Markov random fields, so I can not decide whether the contribution here is significant.

Submitted by Assigned_Reviewer_5

The paper introduces a method to select incrementally a subset of
features that are informative about a specific relevant set of
variables. The issue at hand is about being able to be better than a
generic detector with respect to target variables when these are
specified, and nuisance nodes exist. A core assumption is
Gaussianity which allows much of the procedure to work. Finally, an
enhanced relevance set is introduced to "lift" the original,
non-submodular process towards a sub-modular one which exhibits
performance guarantees for greedy sensor selection.

Though a quite technical paper, I enjoyed reading it. There are lots
of interesting and - to me - novel ideas combined in the work, the
construction of the composable warped MI measure, the relation to
distributed computing, and making the model sub-modular were all
attractive properties of the work. It is a contentful paper, which I
found worth the time reading.

One potential aspect that the reviewer is not sure about is whether
the Gaussian scenario is really relevant in practice for the type of
networks considered here. In particular in large networks with,
possibly, very simple sensors where this approach would be most
interesting, Gaussianity is not really a default, and the question
is how much one would lose when that happens. The authors indicate
that treatment of non-Gaussianity is future work, but a short
discussion on the relevance of the results in case of non-Gaussian
variables (perhaps, as an extreme guiding example, one could
consider binary variables as thought model) may be in place here;
it's clear that a full computation is out of the place here, but
some comments as to what can be expected to change.

Generally clear, notwithstanding its technicality, there are a
number of issues that might be improved in the presentation, I will
list them below:

- line 62: Gaussian belief propagation is introduced as (BP), but
later referred to as GaBP. Please make it consistent.

- line 181: the use of the indicator function seems to miss an
argument - the right side of the inequality is a scalar, the left
side is a function.

- lines 262-264: I read and reread the argument, but it was to
compact. Please rephrase the argument of Remark 6 (first three
lines of the remark).

- line 269: what is a pair of *unique* vertices?

- line 272: the explanation for \zeta is unclear here. How do you
compute it? Where does this term arise from?

- line 289: I did not follow the reasoning why in the term
W(i;j|A) + \zeta(r,j; G), the \zeta contains the r,j pair? In (9),
\zeta is computed for the boundary nodes u and v, but here, j
replaces the boundary node on the opposite side of the path
emanating from r. What if there is a node, say q, further down the
path? Surely \zeta will not be unchanged if we replace j by q?
So, why then, are we allowed to use j? Perhaps it's a simple
argument, but I didn't see it.

- line 322: I did not immediately see the operation suggested in
footnote 2 - I do not see how the setting to -\inf works; what
causes the blocking in the chain following procedure?

- line 399: "running on a fictitious problem" - this formulation is
unfortunate. Do you mean that you generate a surrogate problem ^R
(i.e. \hat R) by making S conditionally independent and then run
your algorithm? This is what I understood. But it may mean
something different. Please clarify.

** Supplementary Material Comments
- line 22: "joint potential over the chain" - please define the
term "joint potential". You do not use jargon in the paper (which
is good), please keep that style in the Supplementary Material.

- line 66: "subchain" -> "subchains"

- line 67: font of I(x_1;x_k)

- lines 69-77: The argument is a bit tight and difficult to
follow. Please reformulate. You have space here.
Summary: Technically strong and interesting paper, many ideas, interesting
tools introduced. Questions about relevance to actual distributed
sensor networks. Some arguments could be expanded/clarified.

Submitted by Assigned_Reviewer_6

This paper proposes a focused active inference problem that selects a subset of observable variables to infer only a subset of latent variables, instead of all the latent variables. By assuming unique path between variables and joint Gaussian distribution, the authors introduced a decomposition of nonlocal mutual information into local information measures that can be efficiently computed using message passing algorithms. Using such a decomposition, they propose an iterative algorithm to select the subset of observable variables to maximize the mutual information with a specified subset of hidden variables, subject to a cost constraint.

Experiment demonstrated the efficiency of the proposed algorithm, especially in large network. In addition, the paper also provided a performance bound on the objective mutual information based on the submodality property of mutual information.

Quality: The active inference problem is interesting and important for many applications. The proposed solution based on nonlocal mutual information decomposition is novel and sound. It is well supported by theoretical analysis.

However, experiment seems a bit lacking as there is no evidence that the focused active learning outperforms unfocused active learning in terms of accuracy, especially when all methods in experiments return the same results.

In addition, experiments should include a comparison between marginalization of nuisance variables and the proposed method to justify authors claim on efficiency.

Clarity: paper is clearly written and well presented.

Originality: Proposed method on nonlocal mutual information is new to the best of my knowledge, even though it only applies to continuous variables and trees. The greedy search algorithm is the same as the existing active sensing algorithms based on sub-modular function.

Significance: Problem is potentially interesting in large scale network and its efficiency improvement is definitely significant over existing methods.

Weaknesses: the assumption that the joint distribution of the sensors and data follow a Gaussian distribution needs be better justified. This assumption may limit the practical utility of this work.
Summary: The proposed method of mutual information decomposition is interesting and novel. But its assumption of Gaussian distribution needs be better justified. The greed sensor selection algorithm is an implementation of the existing approach. The experiments need be further strengthen to support many of authors’ claims.



Submitted by Assigned_Reviewer_7

This paper addresses active learning in trees of Gaussian random variables, where only a "relevant" subset R of latent variables are truly of interest. In the proposed problem setup, a subset A of the observable nodes is to be selected, to maximize the mutual information I(R;A) of the relevant latent variables and the observable nodes; it is assumed there is some cost associated with collecting observations, constrained by an observation budget. The primary technical contribution of this paper is a message passing algorithm (in section 5) which takes advantage of the block structure of the precision matrix of the multivariate Gaussian, and the decomposition of the mutual information presented in section 4.

The paper is clearly organized, and the mathematical background in section 2 lays out the notation used elsewhere in the paper. It is possible that the algorithm in section 5 could be explained in somewhat more detail; in particular, the discussion of runtimes in 285-295 is somewhat brief.

In section 6, the asymptotic efficiency of the inversion-based methods (as clarified by author rebuttal) grows cubicly with the number of nodes. It is evident that the decompositions presented have resulted in a much faster, linear-time algorithm.

Of particular notability here is that the algorithm presented does not actually require a tree structured graph, merely a graph such that nodes of interest are connected by a single unique path. The use of this algorithm in a parallel computing context is also discussed.

Overall, this looks to be an effective, and clearly described, method for active learning in networks of Gaussian random variables.
Summary: This paper presents a decomposition of the mutual information which can be used to perform a greedy sensor selection task in linear time, making it appropriate for use in high-dimensional spaces.
Author Feedback

Author rebuttal: The authors thank the reviewers for their questions, comments, and helpful suggestions for improving the clarity of the paper. Questions raised by the reviews appear to generally focus on the assumptions of Gaussianity, the experimental results, and the clarity of a few technical points concerning nonlocal mutual information decompositions.


One common question in the reviews pertained to the class of distributions considered, which the authors agree was not as clearly motivated in the paper as it could have been. The paper considers the class of Gaussian distributions, and specifically those Markov to tree-shaped undirected graphs, for a number of reasons:

+ The authors view this paper as being foundational to the study of efficient active inference on more complicated (i.e., non-tree/non-Gaussian) continuous distributions, where the efficient information quantification will be related to the Gaussian tree case through approximation bounds. These relationships are the subject of much future work.

+ To the best of the authors' knowledge, the only distributions with additive information distances monotonic in mutual information are tree-shaped multivariate Gaussians and tree-shaped symmetric discrete distributions, the latter requiring a very specific parameterization for the potential functions defined over edges in the factorization of the joint distribution. Treatment of the symmetric discrete case will be added to the supplementary material.

+ The graphical inference community appears to best understand the convergence of message passing algorithms for continuous distributions on subclasses of multivariate Gaussians (e.g., trees, walk-summable models, etc.).

The authors separately reiterate that the performance bounds in Section 7 apply to *any* distribution with nuisances for which there exists a submodular relaxation.


A second set of questions in the reviews pertains to the experimental results of Section 6.

One reviewer suggests a comparison of the accuracy of unfocused and focused selectors for active inference. The purpose of Example 1 is to demonstrate a simple problem for which ignoring the focused nature of an underlying problem can lead to arbitrarily poor performance of an unfocused selector. More broadly, Example 1 is meant to demonstrate a concept of edge information loss; if such losses are ignored by the selector on account of solving only the unfocused problem, one could expect performance to degrade accordingly. So while the performance gap between unfocused and focused selection is strongly problem specific (through the considered precision matrix J, as well as the specified set of [latent] nuisance nodes), Example 1 indicates there is *generally* no bound on the performance loss of an unfocused selector. Future work could develop an analytic tool for determining an *instance-specific* certificate on the performance loss between unfocused and focused selection. If that loss is small, and the unfocused problem is considerably less expensive to solve, the user could decide that unfocused selection will suffice. The paper assumes that the user has decided to solve the focused problem using a focused selector. The authors agree that an empirical comparison of unfocused and focused selection for a subclass of problem instances would be an interesting result, but the conclusions of such a study would be strongly dependent on the distribution from which problem instances are drawn, and a characterization of those problem classes is outside the scope of this paper.

Another suggestion is to include experiments comparing the proposed method to marginalization of nuisance variables. The experiments presented in the paper compare the efficient information quantifier (which uses the mutual information decomposition and GaBP-based algorithm of the paper) with two inversion-based methods. The more efficient of the two inversion-based methods corresponds to a block marginalization of nuisances. Both inversion-based methods have cubic asymptotic time complexity; although this was briefly alluded to in Section 2.2, the authors intend to make this point more clear in the revision.

Finally, the authors make no claims to the originality of doing greedy selection, merely that the paper elucidates a way to perform computationally efficient focused selection.


It is clear from the reviews that the discussion of \zeta in Eq. (9) must be clarified in the paper. For vectoral problems, each node may contain a subnetwork of arbitrarily connected scalar random variables. Under the assumption of thin edges (see Definition 7 and Figure 2b), a unique path between nodes u and v must enter interstitial nodes through one scalar r.v. and leave through one scalar r.v.; the information loss associated with traversing an interstitial vectoral node i is \zeta_i(u,v;G). As node i may have additional neighbors that are not on the u-v path, using the notation \zeta_i(u,v;G) is a convenient way to implicitly specify the enter/exit scalar r.v.s associated with the path. Any unique path subsuming u-v, or any unique path subsumed in u-v for which i is interstitial, will have equivalent \zeta_i terms.